# Evidential Mixture Machines: Deciphering Multi-Label Correlations for Active Learning Sensitivity

**Dayou Yu**[1]    **Minghao Li**[2]    **Weishi Shi**[2]    **Qi Yu**[1]*

Rochester Institute of Technology, Rochester, NY 14623[1]

University of North Texas, Denton, TX 76203[2]

{dy2507,qi.yu}@rit.edu[1]    { minghaoli@my.,weishi.shi@}unt.edu[2]

## Abstract

Multi-label active learning is a crucial yet challenging area in contemporary machine learning, often complicated by a large and sparse label space. This challenge is further exacerbated in active learning scenarios where labeling resources are constrained. Drawing inspiration from existing mixture of Bernoulli models, which efficiently compress the label space into a more manageable weight coefficient space by learning correlated Bernoulli components, we propose a novel model called Evidential Mixture Machines (EMM). Our model leverages mixture components derived from unsupervised learning in the label space and improves prediction accuracy by predicting weight coefficients following the evidential learning paradigm. These coefficients are aggregated as proxy pseudo counts to enhance component offset predictions. The evidential learning approach provides an uncertainty-aware connection between input features and the predicted coefficients and components. Additionally, our method combines evidential uncertainty with predicted label embedding covariances for active sample selection, creating a richer, multi-source uncertainty metric beyond traditional uncertainty scores. Experiments on synthetic datasets show the effectiveness of evidential uncertainty prediction and EMM's capability to capture label correlations through predicted components. Further testing on real-world datasets demonstrates improved performance compared to existing multi-label active learning methods.

## 1   Introduction

Active learning (AL) is a paradigm where we have access to abundant unlabeled data instances with a limited labeling budget [35] [15]. Most AL methods focus on the selection strategy that helps the machine learner achieve better performances with an informed selection of labeled data instances. However, while AL for standard classification tasks has been studied extensively, an important task that is multi-label classification has been largely overlooked. In real-world problems, each data instance may be associated with more than one labels [11, 39, 27]. For machine learning models, the difference between the multi-class problem where only one ground truth label is associated with a data instance and the multi-label classification (MLC) problem is fundamental. Classic solutions either transforming the MLC problem into multiple binary problems or directly build a joint learning problem for all labels. With the binary approach, we lose the common underlying correlations between input features and labels. The number of classifiers may be large due to the large label space, thus the separated training process can be costly. Also, many labels are relatively rare and may depend on other labels, and we can not learn such dependencies when we isolate these labels. With the joint learning approach, the biggest challenge is to combine common labels with rare labels in a balanced

---

*Corresponding author.

38th Conference on Neural Information Processing Systems (NeurIPS 2024).

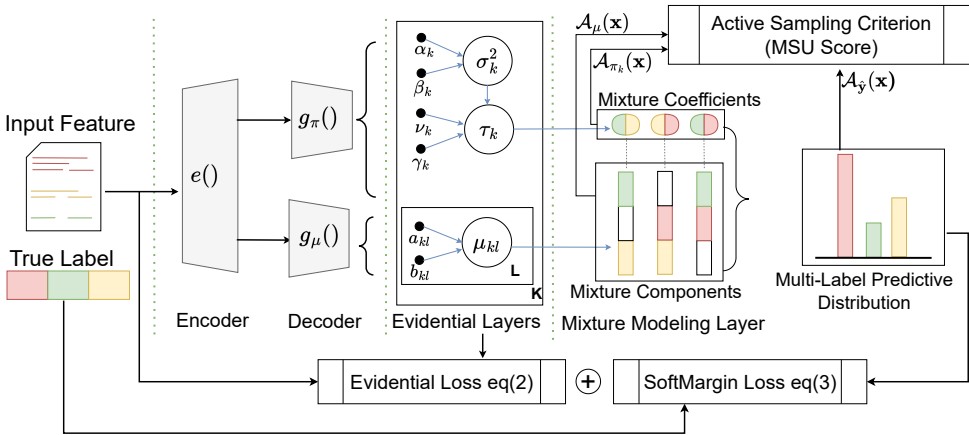

Figure 1: Overview of evidential mixture machines

way. For the rare labels, there might be few positive data instances. A typical model is likely to optimize the feature embedding according to popular labels or adopt the so-called "shortcut" learning, making a direct connection from the input features to the rare labels more difficult. Although we now have access to common and rare labels at the same time, it might not suffice to promote the learning of their correlations if we directly model the labels from the input features. These unique challenges of the MLC problem become more pronounced in AL settings, where the rare labels become even more scarce. They also increase the difficulty of obtaining high-quality uncertainty estimation, which is often crucial in many AL selection strategies as it allows us to know when the model "does not know" in order to make an educated decision on which data instances to label.

To address these challenges, we draw inspiration from the mixture of Bernoulli model, which can model a large label space with a small number of mixture components. There are existing methods that try to capture the label correlation using such a model [17, 30]. However, to connect with the input features, they either resort to a purely conditional case, namely conditional Bernoulli mixtures (CBM) [17] or use a conjugate classification head, which is a Gaussian Process model (GP-B$^2$M) [30]. For the former, a distinct set of label clusters is predicted for each data instance, meaning that the label correlations are completely separated from the learning tasks. This shortcoming makes the CBM model unsuitable for AL. The latter relies on the Gaussian Process (GP) which outputs the weights of the mixture components. For the intended task, a complete GP is too expensive [20, 16], while a sparse GP has limited predictive ability. Unlike the CBM model, GP-B$^2$M creates a set of global label clusters. However, the label prediction for rare labels remains challenging because they can only ever be as good as the best label cluster available. Furthermore, the uncertainty quantification is superficial because it only captures the approximate covariance of the label prediction and the variance of the GP predictions. This point estimate does not fully capture the unknown and is not sufficiently effective in AL.

In this work, we propose to combine the mixture of Bernoulli with a deep evidential model [28], both incorporating deep learning and enabling more fine-grained uncertainty analysis. The deep learning model needs only a forward pass during prediction time, which is much more efficient than estimating the predictive distribution in a random process model. The uncertainty analysis can now include each evidential prediction. This flexibility is crucial in improving the informativeness evaluation of multi-label data samples as mentioned above. Structurally, the proposed model is composed of a shared encoder that provides embeddings and two decoders that predict the weight coefficients and the proxy pseudo counts correction for the Posterior Beta which models the global label cluster, respectively, as shown in Figure 1. The weight coefficient predictor is trained as a deep evidential regression model, which makes uncertainty-aware predictions of optimal assignments to the mixture of label clusters for each data instance. By using a deep evidential model as the classification head, we achieve the aforementioned improvement of efficiency during inference time. By sharing the encoder, we maintain a close connection beginning from the input features, through the weight coefficients, and to the label clusters. Compared to the CBM model, the proposed Evidential Mixture Machines (EMM) maintains a global label cluster. Compared to the GP-B$^2$M model, the EMM model allows

adjustment to the label clusters based on each data instance. Additionally, the uncertainty information can be captured by both the fine-grained uncertainty decomposition from the evidential posterior parameters and the predicted variability in final labels. The evidential uncertainty decomposition is implemented through a conjugate interpretation of the Normal Inverse Gamma parameters, which gives these parameters a more impactful pseudo count meaning. The label variability is assessed using the covariance approximation in predictions and the discrepancy between the global label cluster and the proxy pseudo counts. Apart from the novel sampling strategy, the model is trained using alternating loops of the evidential training step and a joint label-based training step, which strengthens the ability to accommodate individual samples.

With both the demonstration on synthetic datasets and the AL evaluation on real-world datasets, we show the effectiveness of the EMM model. Our main contribution is threefold:

- integration of evidential learning with multi-label classification through EMM,
- principled uncertainty quantification for active sampling using EMM,
- intensive evaluation showing performance gain by EMM for large label spaces, especially on sparse and rare labels.

## 2 Related Works

**Multi-label classification.** In the realm of multi-label classification (MLC) [41] [27], Binary Relevance Models (BRMs) have gained widespread use, leading to the development of various active learning (AL) models based on BRMs. Notable examples include employing the estimated reduction of a BRM loss function as an uncertainty criterion for data sampling, as demonstrated by [38]. A large portion of the multi-label active learning work do not require annotators to label all possible labels for a given data instance [25, 37, 7]. The main consideration behind these approaches is the significant reduction in annotator labeling costs. However, this strategy inevitably breaks the inherent connections between labels, making it impossible to comprehensively measure the informativeness of a data instance using label correlations. Additionally, models designed to handle partially labeled data are required, limiting the applicability of such methods. Therefore, in this paper, we will not compare our approach to these methods. Other approaches integrate the properties of the support vectors of individual support vector machines (SVM) within BRMs, using label correlation more as a means to simplify the querying process than to enhance active sampling [29, 32, 8, 9, 10]. While these methods incorporate label correlation to some extent, such as through label inconsistency [18], label ranking [26], or learning reularization[40], they do not systematically capture label correlations, potentially leading to imprecise uncertainty measures in ML-AL contexts.

**Label correlations and multi-label AL.** Some existing models attempt to explicitly capture label correlations or use latent embeddings to facilitate active multi-label sampling. For instance, the CBM model uses the approximate entropy of predicted labels for data sampling [5], but its dependency on an external multi-class classifier for predicting component coefficients complicates AL due to the challenges in model selection and parameter tuning. Furthermore, CBM, designed primarily for MLC rather than AL, predicts distinct label clusters for each data sample without discovering global label clusters, thus limiting its effectiveness in multi-label AL [17]. Other approaches like correlation-aware method for transfer learning [7] struggle with large and sparse label spaces due to their reliance on kernel functions for measuring label similarity. Compressed sensing (CS) techniques [34, 31] innovative in learning latent embeddings to capture label correlations but assume labels are drawn from a Gaussian distribution and are not efficient in AL, especially in early stages with limited training data. In [30], a Bayesian mixture of Bernoulli model is proposed. A set of global label clusters are captured in a Bayesian manner. However, the inference process of the model is complicated and the fixed label clusters limit the predictive ability in the final label space.

**Evidential learning.** Evidential models have been developed to enable fine-grained uncertainty quantification in deep learning (DL) models [28] [1]. These models introduce a higher order conjugate prior distribution over the likelihood distribution, and train the DL model to output the parameters of the higher order distribution [4] [3]. The higher order distribution enables the model to express the fine-grained uncertainty information. Evidential models have been successfully extended to classification [12, 19], regression [1, 21], action recognition [2], OOD detection [13], and meta-learning problems [22]. We extend the evidential deep learning framework to our setting that leads to novel fine-grained uncertainty guided active-learning for multi-label classification.

# 3 Methodology

**Problem setting.** In our multi-label active learning problem, the essential task is to predict a multi-variate 0-1 label vector $\mathbf{y} = (y_1, ..., y_L)^\top \in \{0,1\}^L$ from the input features $\mathbf{x} \in \mathbb{R}^M$. For AL, we start with a small initial labeled set $\mathcal{S}_L$, and a large unlabeled pool $\mathcal{S}_U$ ($N_L = |\mathcal{S}_L| \ll N_U = |\mathcal{S}_U|$). The AL strategy $\mathcal{A}(\mathbf{x}) : \mathbb{R}^M \to \mathbb{R}$ is a function that grades the instance $\mathbf{x}^* \in \mathcal{S}_U$ according to its informativeness. The top graded samples will be labeled as a batch $b_t$ and added to $\mathcal{S}_L$. The building blocks of our proposed method include the Mixture of Bernoulli label clusters and the coefficient predictor which connects the input features to the label clusters.

## 3.1 Preliminaries

**Mixture of Bernoulli.** The set of label clusters contains $K$ mixture components. Each mixture component is a $L$-variate Bernoulli distribution $\mathbf{z}_k = \prod_{l=1}^L \text{Bernoulli}(y_l; \mu_{kl})$ that captures a local 'stereotype' of the complete label distribution, where $\text{Bernoulli}(y_l; \mu_{kl}) = \mu_{kl}^{y_l}(1 - \mu_{kl})^{(1-y_l)}$ and $L$ is the total number of labels. The Bernoulli parameter $\mu_{kl}$ has conjugate prior $\text{Beta}(\mu_{kl}; a_{kl}, b_{kl})$. Initially, the mixture of Bernoulli can be found through label-only learning. Using an EM algorithm, we can learn the initial components $\boldsymbol{\mu}_{K \times L}^{(0)}$ from the labeled samples $\mathcal{S}_L$. These components can model the set of labels $p(\mathbf{y}|\boldsymbol{\mu}) = \sum_{k=1}^K \pi_k \prod_{l=1}^L \text{Bernoulli}(y_l; \mu_{kl})$, where $\pi_k \in (0,1]$ is the normalized weight coefficient of the component $k$. However, since this is a label-only process, we are missing the connection to the input features $\mathbf{x}$.

**Connecting to input features.** For a conditional model such as CBM, the connection is through $\pi_k = p(\mathbf{z}_k|\mathbf{x})$ and $\mu = \mu(\mathbf{x})$. The model is still trained in an EM manner so that the predictions $\hat{\mathbf{y}}$ can be made for each $\mathbf{x}$. For a conjugate model such as GP-B$^2$M, the connection is through $\pi_k = p(\mathbf{z}_k|\text{GP}_k(\mathbf{x}))$, and the variational training process of the conjugate model impacting the posterior $\boldsymbol{\mu}$. As mentioned before, one issue with CBM is the disconnected prediction models of $\boldsymbol{\pi}$ and $\boldsymbol{\mu}$, while the inference process of GP-B$^2$M is too expensive. In our proposed model, we also predict both $\hat{\boldsymbol{\pi}}$ and $\hat{\boldsymbol{\mu}}$ from $\mathbf{x}$ using two decoder networks $g_{\boldsymbol{\pi}}(\cdot)$ and $g_{\boldsymbol{\mu}}(\cdot)$. However, we maintain the connection by using a shared encoder network $e(\mathbf{x})$, as shown in the structure in Figure 1. Specifically, $g_{\boldsymbol{\pi}}(e(\mathbf{x}))$ is an evidential model that predicts the distribution of $\boldsymbol{\pi}$ using the output evidence parameters, which function in the following way.

**Evidential weight coefficient predictor.** We model the connection from input features to label clusters with fine-grained uncertainty information using the evidential regression model. The target for regression is to learn a combination of $\pi_k$ values that best reconstruct the final label predictions. To this end, we place a higher-order Normal Inverse Gamma (NIG) prior $\text{NIG}(\tau, \sigma^2|\mathbf{p}) = \mathcal{N}(\tau|\gamma, \frac{\sigma^2}{\nu})\Gamma^{-1}(\sigma^2|\alpha, \beta)$ over the regression model's Gaussian output $\mathcal{N}(\pi|\tau, \sigma)$. The evidential model is trained to output the NIG parameters $\mathbf{p} = (\gamma, \nu, \alpha, \beta)$ similar to [1]. In this evidential model, the Gaussian likelihood interacts with the NIG prior, leading to a Student-t predictive distribution:

$$p(\pi|x, \mathbf{p}) = \int_\tau \int_{\sigma^2} p(\pi|x, \tau, \sigma^2)\text{NIG}(\tau, \sigma^2|\mathbf{p})\mathrm{d}\tau\mathrm{d}\sigma^2 = \text{St}\left(\pi; \gamma, \frac{\beta(1+v)}{v\alpha}, 2\alpha\right) \tag{1}$$

Here, the evidential model predicts coefficients for input $x$ as: $\hat{\pi} = \mathbb{E}_{p(\pi|x,\mathbf{p})}[\pi] = \gamma$.

The NIG parameters $\gamma, \nu, \alpha, \beta$ are outputs from the predictor branch $g_{\boldsymbol{\pi}}(e(\mathbf{x}))$. The evidential model, through its higher order NIG prior, can quantify the aleatoric (ALE) and epistemic (EP) uncertainty [1] as $\text{ALE} = \mathbb{E}[\sigma^2] = \frac{\beta}{\alpha-1}$, $\text{EP} = \text{Var}[\tau] = \frac{\beta}{\nu(\alpha-1)}$. In this evidential framework, due to the conjugacy of the NIG prior with the Gaussian likelihood, the posterior is also the NIG distribution. Moreover, in this model, after interacting with $N$ i.i.d. data points $(\boldsymbol{\pi}_1, ..., \boldsymbol{\pi}_N)$, the posterior NIG parameters update as the observations increase [23]. Through the pseudo-count interpretation, the total evidence is quantified as $\mathcal{E} = v + \frac{1}{2}\alpha + \frac{1}{\beta}$.

We train the model to maximize The likelihood under the predictive Student-t distribution, which gives the NLL loss:

$$\mathcal{L}_{\text{NLL}} = -\log(p(\pi_k|x, \mathbf{p})$$
$$= \frac{1}{2}\log\left(\frac{\pi}{\nu}\right) - \alpha\log\Omega + \left(\alpha + \frac{1}{2}\right)\log\left((\pi_k - \gamma)^2\nu + \Omega\right) + \log\left(\frac{\Gamma(\alpha)}{\Gamma(\alpha + \frac{1}{2})}\right)$$

where $\Omega = 2\beta(1 + \nu)$. Additionally, we want the model's confidence/evidence for the prediction to be low when the prediction is incorrect by introducing a evidence-based regularization $\mathcal{L}_{REG} = (\pi_k - \gamma)^2 \cdot \mathcal{E}$. The regularization penalizes the highly confident wrong predictions, and ensures model's confidence is rightly placed. The overall loss is

$$\mathcal{L}_{\texttt{EVID}} = \mathcal{L}_{\texttt{NLL}} + \lambda_{reg}\mathcal{L}_{\texttt{REG}} \tag{2}$$

where $\lambda$ controls the impact of the regularization.

### 3.2 Bringing Evidential Learning into Multi-Label Active Learning

The integration of an evidential model in our multi-label classification approach presents several distinct advantages, particularly in addressing the inherent complexities of active learning (AL) environments. Firstly, evidential models provide a more nuanced and sophisticated mechanism for uncertainty quantification. This is crucial in AL settings, especially when dealing with sparse and rare labels, where traditional models often struggle. By effectively capturing and quantifying uncertainty, our approach leads to more informed and strategic decisions selecting data instances for labeling, optimizing the use of limited labeling resources. Furthermore, the evidential model facilitates a deeper understanding of the underlying label correlations, enabling the model to make more accurate predictions across a broad spectrum of labels, including the rare ones. This leads to a significant improvement in the overall classification performance, especially in scenarios where conventional methods might overlook subtle but crucial label dependencies. Additionally, the evidential approach inherently enhances the interpretability of the model's predictions, offering insights into the confidence and reliability of these predictions. This aspect is particularly valuable in knowledge-rich domains where understanding the model's decision-making process is as important as the predictions accuracy. Thus, incorporating an evidential model into our multi-label classification framework marks a substantial advancement, offering a robust, efficient, and insightful solution to the challenges posed by large and complex label spaces in active learning scenarios.

### 3.3 Evidential Mixture Machines

Building upon the tools above, we propose a novel EMM model. Compared to existing methods, our novel contribution to model learning lies in how we connect label clusters to input features and how we learns the final labels in a joint manner. The integration enables evidential uncertainty analysis through both weight coefficient predictions and final label predictions. The evidential learning of $\boldsymbol{\pi}$ is already explained above. Here, we introduce the joint learning using actual labels $\mathbf{y}$.

**Joint multi-label training with label clusters.** Once we have the coefficient predictor branch, we can train the full model to make the final label predictions. We first freeze $e(\cdot)$ and $g_{\boldsymbol{\pi}_k}(\cdot)$ to train $g_{\boldsymbol{\mu}}(\cdot)$. Instead of directly letting the network predict $\boldsymbol{\mu}$ from $g_{\boldsymbol{\mu}}(\cdot)$, we make $g_{\boldsymbol{\mu}}(\cdot)$ output proxy pseudo counts $\hat{a}_{kl}$ and $\hat{b}_{kl}$ to be combined with the initial $(\boldsymbol{\mu}_{K \times L}^{(0)}; a_{kl}^{(0)}, b_{kl}^{(0)})$. This ensures that we maintain the correlations encoded in $\boldsymbol{\mu}_{K \times L}^{(0)}$. The network outputs of dimension $2 \cdot N \cdot K \cdot L$ are split into $\hat{a}$ and $\hat{b}$ and added to $a^{(0)}$ and $b^{(0)}$ in a weighted fashion: $a_{kl}(\mathbf{x}) = a_{kl}^{(0)} + w_\mu \hat{a}_{kl}(\mathbf{x}), b_{kl}(\mathbf{x}) = b_{kl}^{(0)} + w_\mu \hat{b}_{kl}(\mathbf{x})$. The new Bernoulli parameter for each instance is then computed by $\mu_{kl}(\mathbf{x}) = a_{kl}(\mathbf{x})/(a_{kl}(\mathbf{x}) + b_{kl}(\mathbf{x}))$. The label prediction is $\hat{y}_l(\mathbf{x}) = \sum_k \pi_k(\mathbf{x})\mu_{kl}(\mathbf{x})$. The model is trained using a soft margin multi-label loss:

$$\mathcal{L}_{\texttt{SoftMargin}} = -\frac{1}{L}\sum_{l=1}^{L} y_l \log\left(\frac{1}{1 + \exp(-\hat{y}_l)}\right) + (1 - y_l)\log\left(\frac{\exp(-\hat{y}_l)}{1 + \exp(-\hat{y}_l)}\right) \tag{3}$$

We then adopt the evidential pseudo-count style update of the label clusters, we would have $a_{kl}^{(1)} = a_{kl}^{(0)} + \sum_{k=1}^{K} \hat{\pi}_k(\mathbf{x}_n)y_{nl}$ and $b_{kl}^{(1)} = b_{kl}^{(0)} + \sum_{k=1}^{K} \hat{\pi}_k(\mathbf{x}_n)(1 - y_{nl})$. However, this update only makes the correction based on the predicted weights $\hat{\mu}_k$. If we keep updating in this way, the biases build up and the popular labels will be dominant in future components. Thus, we also include a weighted update based on the predicted proxy pseudo-counts, similar to when we make label predictions: $a_{kl}(\mathbf{x}) = a_{kl}^{(0)} + w_{up}\hat{a}_{kl}(\mathbf{x}), b_{kl}(\mathbf{x}) = b_{kl}^{(0)} + w_{up}\hat{b}_{kl}(\mathbf{x})$. This step ensures that our model mutually benefits from the coefficient predictor and the proxy pseudo-count predictor. The initial components are learned unsupervised and do not make up for the training of the predictors. By introducing the joint update, we connect the two predictors more closely.

The joint training step of EMM addresses an important problem with the mixture model formulation, where the model prediction is restricted by the mixture components $\boldsymbol{\mu}$. Because the weight coefficients

$0 \leq \pi_k \leq 1$, the label prediction for $y_l$ can only be as great as $\max_k \mu_{kl}$. If we only make updates to the mixture components using $a_{kl}^{(1)} = a_{kl}^{(0)} + \sum_{k=1}^{K} \hat{\pi}_k(\mathbf{x}_n) y_{nl}$ and $b_{kl}^{(1)} = b_{kl}^{(0)} + \sum_{k=1}^{K} \hat{\pi}_k(\mathbf{x}_n)(1 - y_{nl})$, the rare labels will still suffer from the label imbalance which will be reflected in $\max_k \mu_{kl} = \max_k \frac{a_{kl}}{a_{kl}+b_{kl}}$. By incorporating the joint training, we make the prediction based on $a_{kl}(\mathbf{x}) = a_{kl}^{(0)} + w_\mu \hat{a}_{kl}(\mathbf{x}), b_{kl}(\mathbf{x}) = b_{kl}^{(0)} + w_\mu \hat{b}_{kl}(\mathbf{x})$, allowing the model to better fit the labels using instance-wise predictions $\hat{a}_{kl}(\mathbf{x}), \hat{b}_{kl}(\mathbf{x})$. The soft margin multi-label loss effectively brings the benefits of binary relevance machines into model training because it promotes positive predictions through the multi-label one-versus-rest formulation.

**Complete EMM learning process.** Having established the weight coefficient training step and the joint multi-label training step, we integrate them in a complete learning process. To start with, we have the EM-learned initial clusters $\boldsymbol{\mu}_{K \times L}^{(0)}$ and the optimal $\Pi^{(0)}$ which reconstructs the labels when combined with $\boldsymbol{\mu}_{K \times L}^{(0)}$. In one learning round, the model first goes through a pre-training stage of $g_{\boldsymbol{\pi}}(e(\mathbf{x}))$ to fit the set of $\Pi^{(0)}$ optimized for the initial $\boldsymbol{\mu}_{K \times L}^{(0)}$. Then, we move on to the joint training stage where we alternate between training the coefficient predictor and jointly training the full model to fit the labels $\mathbf{y}$. Each coefficient predictor training step is the same as the pre-training stage, while the joint training step is described above. The model will continually improve the quality of label clusters to maximize the ability to recreate the labels in the joint training step, each time followed by the evidential learning of optimal $\boldsymbol{\pi}$ to keep up with the new label clusters. The complete training process is presented in Figure 1. At this stage, we have combined the advantages of Bayesian mixture models, deep evidential models, and a bi-level multi-label problem formulation to obtain a powerful multi-label classification model. Next, we introduce how the evidential flavor can provide fine-grained uncertainty information and benefit active learning.

### 3.4 Active Learning Strategy

In order to select the most informative samples given the small initial budget, we adopt an uncertainty-oriented selection strategy. From the proposed EMM model, we can obtain uncertainty information from three sources: weight coefficient branch, proxy pseudo count predictor, and the final label predictions. The first part of the uncertainty information is from the evidential model that predicts the weight coefficients. The evidential model naturally decomposes into the aleatoric uncertainty $\mathbb{E}[\sigma_{\pi_k}^2]$ and the epistemic uncertainty EP. For AL purposes, we should target the samples that give us the most epistemic uncertainty, which can potentially improve the model's knowledge of the unknown. From the weight coefficient perspective, this criterion searches for the least confident samples based on our current model. Selecting these samples will help us quickly gain knowledge of the connection between input features and the weight coefficients, which is the determining factor for predictive performance. Thus, the first part of the selection function is

$$\mathcal{A}_{\pi_k}(\mathbf{x}) = \frac{\beta_k(\mathbf{x})}{\nu_k(\mathbf{x})(\alpha_k(\mathbf{x}) - 1)} \tag{4}$$

The second part of the uncertainty information comes from the proxy pseudo counts. We can compare the current components and the updated components when the proxy pseudo counts for $\mathbf{x}$ are added, and select the samples that would introduce more difference to the current model: $\mathcal{A}_\mu(\mathbf{x}) = -\text{CosineSimilarity}(\boldsymbol{\mu}, \boldsymbol{\mu}'(\mathbf{x})) = \frac{\boldsymbol{\mu} \cdot \boldsymbol{\mu}'(\mathbf{x})}{\|\boldsymbol{\mu}\| \cdot \|\boldsymbol{\mu}'(\mathbf{x})\|}$. Because the label clusters play the most important role in recreating the label space, we should try to capture as much latent label correlation as possible. This requires sufficient exploration in the label space. Selection criterion $\mathcal{A}_\mu(\mathbf{x})$ does exactly this by searching for data samples that are potentially the most different from the currently captured label space. The last part of the uncertainty information is computed over the final label prediction. Since the full model is a mixture of Bernoulli, we can compute the expected covariance of the predicted label distribution. Here, we adopt the point estimate as in existing methods:

$$\text{cov}[\hat{\mathbf{y}}|\mathbf{x}] = \sum_k \pi_k \left( \text{diag}(\boldsymbol{\mu}(1-\boldsymbol{\mu})) + \boldsymbol{\mu}_k \boldsymbol{\mu}_k^\top \right) - \mathbf{p}(\hat{\mathbf{y}}|\mathbf{x}) \mathbf{p}(\hat{\mathbf{y}}|\mathbf{x})^\top \tag{5}$$

Note that ideally we would like to use the conditional entropy $H[y|x]$ to measure the uncertainty raised due to observing input $x$. However, the entropy of a mixture random variable is hard (or intractable) to compute because of the sum in the expression. But under mild conditions, we can show that $y$, as an average of the combination of weights and components, converge to a normal

distribution so that the $cov(y|x)$, which is easy to compute, can be used to quantify the uncertainties in the mixture random variables.

Let $\phi_1, ..\phi_K$ be the i.i.d samples that encapsulates the coefficient-component pairs where $\phi_i = \begin{pmatrix} \pi_i\mu_{1i} \\ ... \\ \pi_i\mu_{Li} \end{pmatrix}$ with mean $\mathbf{m} = \begin{pmatrix} m_1 \\ ... \\ m_l \end{pmatrix} = \begin{pmatrix} \mathbb{E}[\pi_i\mu_{1i}] \\ ... \\ \mathbb{E}[\pi_i\mu_{Li}] \end{pmatrix}$ and covariance $\Sigma$. Let $y_{|x} = \begin{pmatrix} \overline{y}_1 \\ ... \\ \overline{y}_L \end{pmatrix}$ where $\overline{y}_L = \sum_{k=1}^{K} \pi_k\mu_{kl}$. Then, according to the Multivariate Central Limit Theorem, we have $y \sim \mathbb{N}(\frac{\mathbf{m}}{K}, \frac{\Sigma}{K})$. So we can leverage the dominant term in the entropy of multivariant Gaussian, $\ln(|cov(y|x)|)$ as a surrogate measurement of $H[y|x]$, since the number of components is relatively large.

The selection score is $\mathcal{A}_{\hat{\mathbf{y}}}(\mathbf{x}) = \log |\text{cov}[\hat{\mathbf{y}}|\mathbf{x}]|$. This criterion captures the expected information gain evaluated on the final label predictions when including the unlabeled samples. It aggregates the predictions from the weight coefficient predictor and the proxy pseudo-count predictor, and shapes the fine-grained uncertainty in a global view. By focusing on epistemic uncertainty from the evidential model for weight coefficients, differences in label clusters indicated by proxy pseudo counts, and the covariance in label predictions, we devise a comprehensive multi-source uncertainty-based selection score (MSU) $\mathcal{A}(\mathbf{x}) = \mathcal{A}_{\pi_k}(\mathbf{x}) + \lambda\mathcal{A}_\mu(\mathbf{x}) + \eta\mathcal{A}_{\hat{\mathbf{y}}}(\mathbf{x})$. Compared to a single uncertainty score, the integration of these uncertainty measures facilitates a targeted exploration of the data space and enables the identification of the most informative samples within a constrained budget.

# 4 Experiments

We conduct experiments on both synthetic and real data to demonstrate the effectiveness of EMM. We use the synthetic data experiment to show how label clusters of the mixture model can capture various label compositions and correlations. We then verify the AUC performance of EMM on real-world multi-label datasets, along with rare label analysis and ablation studies.

## 4.1 Synthetic Data Experiments

To demonstrate the effectiveness of the EMM model and the MSU sampling strategy, we design a synthetic dataset that can verify each of the proposed functionalities. The synthetic dataset contains mostly geometric feature related labels, along with a few carefully designed labels. The input features of the data points consist of 16 clusters distributed as $m$-dimensional Gaussian's. In other words, each point is sampled from one Gaussian cluster. The clusters have different means and universal variances such that they have slight

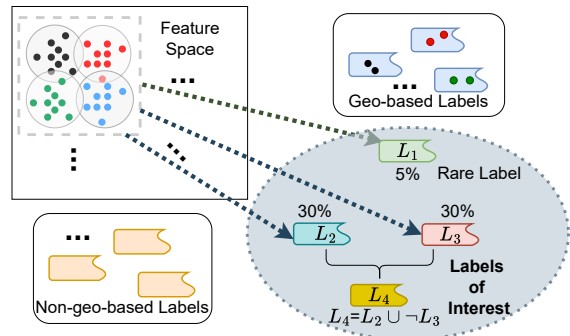

Figure 2: Visualization of label composition in the synthetic dataset. For visualization we show the geometric clusters in 2D, while they are high dimensional Gaussians in practice.

overlaps. The geometric feature related labels indicate which Gaussian the point is sampled from. We denote all the geometric-related labels as $\mathbf{L}_{geo}$'s (all 16 of them) for simplicity. While $\mathbf{L}_{geo}$'s mainly indicate the location of the data samples in feature space and do not carry correlations in themselves, we construct the *labels of interest* using certain groups of them to ensure complex label correlations that are also feature-rooted. These *labels of interest* include a rare label **L1**, which has a frequency as low as 5% of a regular label; a couple of highly-correlated labels **L2** and **L3** which share similar features; and **L4**, which only depends on **L2** and **L3** and is dependent on other labels instead of input features. Specifically, **L1** is assigned to data samples randomly with a low probability of 5%. **L2** and **L3** are randomly sampled from the same quarter of the feature space. Then, **L4** is generated following the rule **L4**= **L2** $\cup$ ¬**L3**. Besides these labels, we also append a set of non-geometry information-based labels $\mathbf{L}_{non-geo}$, which prevent the problem from being purely geometry-based. The label composition is shown in Figure 2.

**Capturing label correlations.** In one example of our experiments, we train the EMM with 6 label clusters. Among these clusters, one has a particularly high weight $\mu_{\{1,L2\}}$ for **L2**. Simultaneously, the weight $\mu_{\{1,L3\}}$ for **L3** is low while the weight $\mu_{\{1,L4\}}$ for **L4** is high. Such behavior will ensure that the co-appearing **L2** and **L4** are captured during the prediction process. In the 6-cluster setting,

Table 1: The relationship between average uncertainty scores, label cardinality, and rare labels

|  | $|\mathbf{y}|<3$ | $|\mathbf{y}|\geq 3$ | $\mathbf{y}_{L1}=1$ | $\mathbf{y}_{L1}=0$ |
|---|---|---|---|---|
| Average $\mathcal{A}_{\pi_k}(\mathbf{x})$ | 46.5 | 36.1 | 52.9 | 31.9 |
| Average $\mathcal{A}_{\hat{\mathbf{y}}}(\mathbf{x})$ | 0.079 | 0.070 | 0.138 | 0.069 |

only $\boldsymbol{\mu}_{\{1,.\}}$ and $\boldsymbol{\mu}_{\{3,.\}}$ have higher weights for **L2**, and in both clusters $\boldsymbol{\mu}_{\{.,L4\}}$ is similarly high as $\boldsymbol{\mu}_{\{.,L2\}}$. Meanwhile, in $\boldsymbol{\mu}_{\{6,.\}}$ we have a high weight $\mu_{\{3,L3\}}$ for **L3**, in which case $\mu_{\{3,L4\}}$ for **L4** is low (« 0.01). Furthermore, in cluster 4 $\boldsymbol{\mu}_{\{4,.\}}$, both $\mu_{\{4,L2\}}$ and $\mu_{\{4,L3\}}$ have slightly higher values but $\mu_{\{4,L4\}}$ is low (« 0.01) so **L4** does not falsely appear.

We further test various numbers of components and quantify the label correlations described in those label clusters. The results show that the average $\text{CosineSimilarity}(\mu_{\{.,L2\}}, \mu_{\{.,L4\}})$ is 0.91 meaning that the positive correlation between **L2** and **L4** is always captured. Meanwhile, the average $\text{sim}(\mu_{\{.,L3\}}, \mu_{\{.,L4\}})$ is as low as 0.12 showing the lack of correlation between **L3** and **L4**. More specifically, when neither $\mu_{\{.,L2\}}$ or $\mu_{\{.,L3\}}$ is insignificant, $R(\mu_{\{.,L2\}}, \mu_{\{.,L4\}})$ drops to 0.31, indicating that the fine relationship of "if and only if" is well-captured by the mixture model.

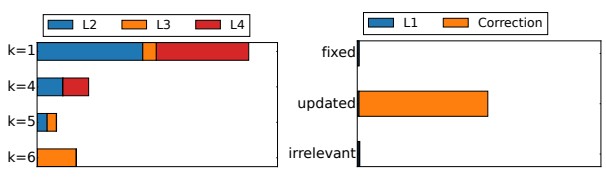

(a) Clusters with **L2**, **L3**, **L4**  (b) Clusters with **L1**

Figure 3: (a) A visualization of the labels clusters concerning **L2**, **L3**, and **L4**; (b) A visualization of the labels clusters concerning **L1** with and without updating with proxy pseudo-counts. "fixed" is the original unsupervisedly trained $\mu_{1,L1}$, "updated" is an updated $\mu_{1,L1}(\mathbf{x}_1)$ where $\pi_1(\mathbf{x}_1) = 0.83$ meaning $\boldsymbol{\mu}_{1,.}$ is dominant, and "irrelevant" is an updated $\mu_{1,L1}(\mathbf{x}_2)$ where $\pi_1(\mathbf{x}_2) = 0.18$ meaning this cluster $\boldsymbol{\mu}_{1,.}$ does not contribute much to the prediction of $\mathbf{x}_2$.

**Prediction enhancement by proxy pseudo-count combination.** Although the mixture model is great at capturing the label correlations with label clusters, it is not always good at predicting rare labels. For example, in the 6-cluster setting, the largest weight for **L1** is $\mu_{\{2,L1\}} = 0.016$ because of the extremely imbalanced label distribution. In this case, even if the model predicts solely $\boldsymbol{\mu}_{\{2,.\}}$ for a sample $\mathbf{x}$ ($\pi_2 = 1, \pi_k = 0, k \neq 2$), the prediction of **L1** is 0.016. This small value creates difficulty in converting the predicted score to a positive label prediction.

**Rare label and correlation discovery by uncertainty quantification.** As for actively selecting data samples, we study the correlations between $\mathcal{A}_{\pi_k}(\mathbf{x})$, $\mathcal{A}_{\hat{\mathbf{y}}}(\mathbf{x})$, and the true labels of $\mathbf{x}$. We show a set of statistics in Table 1 to analyze these behaviors. For $\mathcal{A}_{\pi_k}(\mathbf{x})$, we compare the sampling score with the estimated unknown information of the corresponding pool samples. The unknown information is evaluated from both the feature and label perspective, using the feature similarity and the label cardinality. The correlation between the feature similarity and the uncertainty score is -0.73, and the correlation between the label similarity and the uncertainty score is -0.41. From the label similarity we can also conclude the negative correlation between the similarity and uncertainty. For $\mathcal{A}_{\hat{\mathbf{y}}}(\mathbf{x})$, we specifically focus on the rare labels. On average, the samples containing less than 3 labels have an uncertainty score 28.8% higher than the other samples and the samples containing rare labels have an uncertainty score 65.8% higher than the regular samples.

## 4.2 Real Data Experiments

**Datasets and experiment settings.** We conduct AL experiments on representative real-world datasets including Delicious, Bibtex, Corel5k, Enron, and NUS-WIDE, covering multiple application domains [33, 6]. The number of labels ranges from 53 to 156, most of which are relatively rare in the entire dataset. We summarize the datasets and data preprocessing in Appendix D.

**Performance comparison.** We compare the AL performance with competitive AL baselines:

- **GP-B²M** uses a Bayesian mixture model and conducts active sampling using the combined predicted variance of multi-output GP and the label clusters [30].
- **MMC** is model-adaptive (implemented with label ranking model) and involves a predictive process for the number of labels. It samples instances based on the expected loss [38].
- **Adaptive** Adaptive uses SVM margin and label cardinality inconsistency for data sampling. [18].

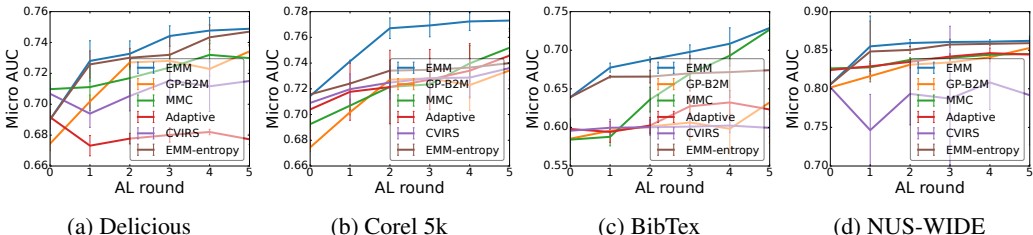

| (a) Delicious | (b) Corel 5k | (c) BibTex | (d) NUS-WIDE |

Figure 4: Performances on real-world datasets (AU-ROC increases as we sample 5 rounds with 100 samples in each round)

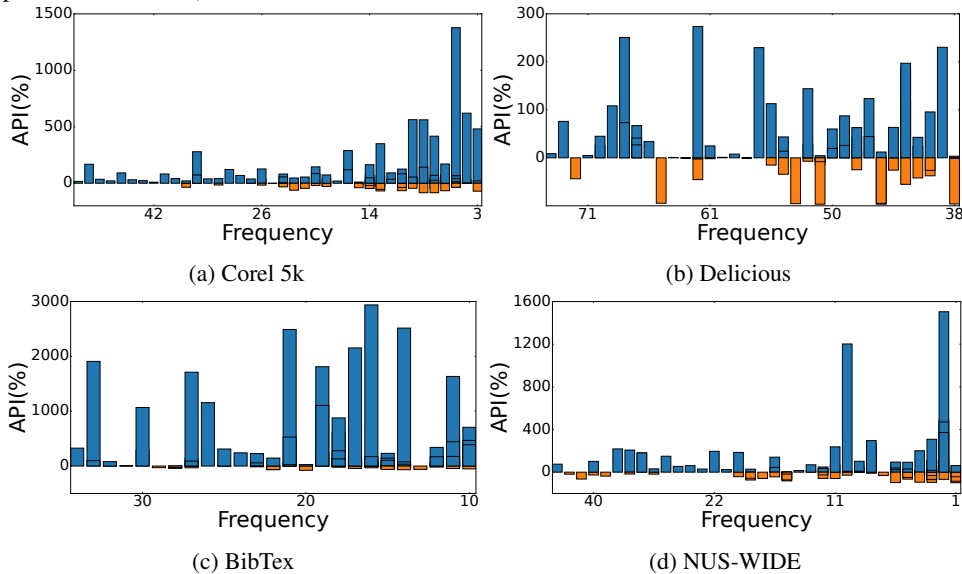

| (a) Corel 5k | (b) Delicious |

| (c) BibTex | (d) NUS-WIDE |

Figure 5: Average precision improvement (API) for rare labels

- **CVIRS** combines the margin based margin ranking and label inconsistency for data sampling [26].

For AL comparison, we use the area under the ROC curve as the main criterion. We start with $2\%$ initially labeled samples for datasets Delicious, Bibtex, Corel5K, and $0.03\%$ for NUS-WIDE. The initial labeled set contains a minimum of one positive instance per label to ensure that binary solutions can be trained. For EMM and methods that can perform batch active learning, we sample 5 rounds with 100 samples selected in each round. For single-batch baseline methods, we sample 500 rounds to obtain the same number of labels in the end. The base performance of classification models varies as some baseline methods use binary-SVMs (Adaptive, CVIRS), some use strategy-specific models such as the label-ranking model (MMC) and the GP-B$^2$M model.

We also include a configuration EMM-entropy that uses the proposed EMM model and a simple entropy-based selection strategy as an additional baseline, showcasing the performance gain from the proposed sampling (MSU selection) on its own. From Figure 4, we can see that the EMM model makes better predictions using the same amount of initial labels compared to the SVM model, which explains the advantage at the starting point. Although the label ranking model or the GP-B$^2$M model may also have good performance at the starting point, they are restricted by specific sampling methods. To separately verify the advantage brought by the uncertainty quantification, we show that our MSU selection always has an advantage in selecting better AL samples compared to a simple uncertainty-based selection (EMM-Entropy).

For a more fine-grained analysis of the model performance, we also compute the average precision improvement [30]. This shows how the rare-label predictions have improved using the EMM model compared to a fully Bayesian mixture model where the label clusters are completely global.

$$\text{API}_l(\%) = \frac{\text{AP}_l(\textbf{EMM}) - \text{AP}_l(\textbf{GP-B}^2\textbf{M})}{\text{AP}_l(\textbf{GP-B}^2\textbf{M})} \times 100\% \qquad (6)$$

In Figure 5, we show the API metric for the 50 rarest labels on each dataset. The improved API on a label is shown by a blue bar above the $API = 0$ axis, while the worse API performances are shown

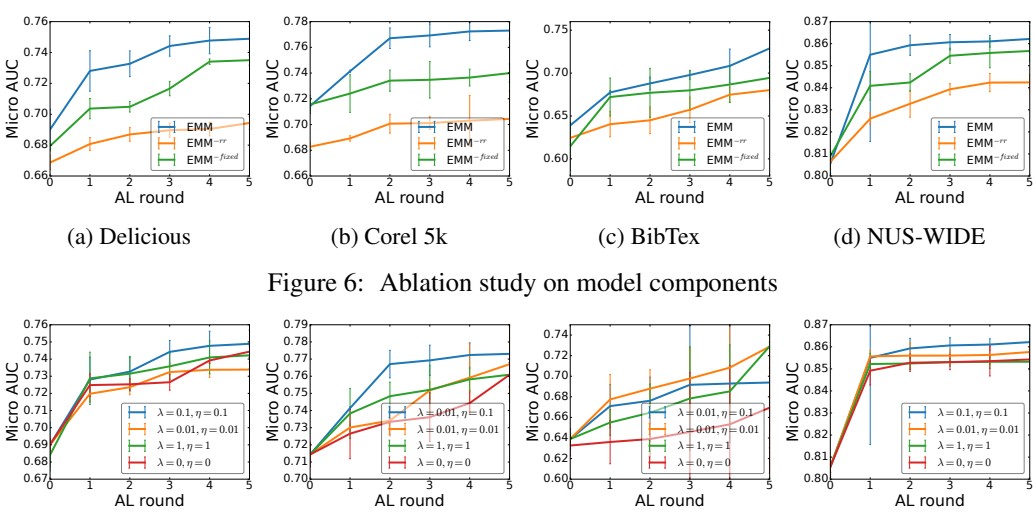

Figure 6: Ablation study on model components

|  |  |  |  |
|---|---|---|---|
| (a) Delicious | (b) Corel 5k | (c) BibTex | (d) NUS-WIDE |

Figure 7: Ablation study on balancing parameters

|  |  |  |  |
|---|---|---|---|
| (a) Delicious | (b) Corel 5k | (c) BibTex | (d) NUS-WIDE |

by the orange bars. The $x$ axis shows the number of times each label has appeared in the testing set. We can see that EMM has a significant advantage on rarer labels.

**Ablation study.** We conduct an ablation study on model components (weight coefficient and proxy pseudo-count predictors) and the AL sampling method (balancing parameters $\lambda$ and $\eta$). Since the proposed EMM model combines the evidential weight coefficient learning and the proxy pseudo-counts learning, we compare the complete model with two weakened configurations:

- **EMM**$^{-rr}$ reduces the weight coefficient leaner to a simple ridge regression model, and only combines the prediction with fixed Bernoulli mixtures as the label clusters.
- **EMM**$^{-fixed}$ uses the evidential learning of weight coefficients with only the fixed Bernoulli mixtures as the label clusters.

From Figure 6, we can see that the evidential regression model predicts the weight coefficients better than a simple regression model such as Ridge Regression, which shows the effectiveness of the first branch of EMM ($e()$ and $g_{\boldsymbol{\pi}}()$). We can also see that without the proxy pseudo-count updates, the performance is not as good, which shows the effectiveness of the second branch of EMM ($g_{\boldsymbol{\mu}}()$) and the joint training of the entire model.

From Figure 4, we can already see that the proposed evidential uncertainty-based sampling outperforms a simple metric such as Entropy. From Figure 7, we can see that our MSU sampling strategy effectively benefited from the multi-source uncertainty information compared to the single-source uncertainty ($\lambda = \eta = 0$). However, the epistemic uncertainty from the evidential model is still the most important source of uncertainty as the sampling performance decreases when we increase the balancing parameters for $\mathcal{A}_{\mu}(\mathbf{x})$ and $\mathcal{A}_{\hat{\mathbf{y}}}(\mathbf{x})$ too much.

## 5 Conclusion

In this work, we introduced a novel Evidential Mixture Machines (EMM) model, which integrates deep evidential learning with a multi-label classification framework of AL. This approach effectively tackle a large and sparse label space, particularly addressing the challenges posed by sparse and rare labels. Our model's sophisticated uncertainty quantification and improved prediction accuracy set it apart from traditional Binary Relevance Models and other existing methodologies. The effectiveness of the EMM model is demonstrated through rigorous evaluations on both synthetic and real-world datasets, showcasing its superiority in diverse labeling scenarios. This advancement not only contributes to the development of more efficient MLC methods but also paves the way for future research in this domain. The potential for scaling this approach to larger datasets and adapting it to various domains offers exciting opportunities for further exploration and refinement.

## Acknowledgments

This research was supported in part by an NSF IIS award IIS-1814450. The views and conclusions contained in this paper are those of the authors and should not be interpreted as representing any funding agency. We would like to thank the anonymous reviewers for their constructive comments.

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

# Appendix

**Organization.** In this appendix, we provide additional details of the work, including a summary of notations in Appendix A, a detailed evidential model interpretation in Appendix B, detailed algorithm blocks C, and the experiment details with additional results in Appendix D.

## A   Summary of Notations

Table 2:  Summary of key notations with definitions

| Notation | Definition |
|---|---|
| $\mathbf{y} = (y_1, ..., y_L)^\top$ | Multi-label vector |
| $\mathbf{x}$ | Data feature vector |
| $\mathbf{z}_k = \prod_{l=1}^{L} \text{Bernoulli}(y_l, \mu_{kl})$ | L-variate Bernoulli random variable. |
| $\mu_{kl}$ | Beta random variable |
| $(a_{kl}, b_{kl})$ | Parameters of the Beta distribution. |
| $\pi_k$ | Gaussian random variable (component coefficient). |
| $\mathbf{p} = (\gamma, \nu, \alpha, \beta)$ | Parameters of inverse normal gamma distribution. |
| $\mathcal{N}(\tau\|\gamma, \frac{\sigma^2}{v})$ | Gaussian distribution in the evidential NIG prior with mean $\gamma$ and variance of $\frac{\sigma^2}{v}$ |
| $\Gamma^{-1}(\sigma^2\|\alpha, \beta)$ | Inverse Gamma distribution in the evidential NIG prior with shape parameter $\alpha$ and scale parameter of $\beta$ |
| $St(\pi; \cdot, \cdot, \cdot)$ | Student t distribution |
| Evidence($\mathcal{E}$) | The integrated model evidence |
| $\mathcal{A}(\mathbf{x})$ | Active sampling score function |

## B   Detailed Evidential Model Interpretation

In this section, we introduce the details of the evidential regression model that predicts $\boldsymbol{\pi}$. Specifically, the Student-t predictive distribution is obtained through:

$$p(\pi|x, \mathbf{p}) = \int_\tau \int_{\sigma^2} p(\pi|x, \tau, \sigma^2)\text{NIG}(\tau, \sigma^2|\mathbf{p})\mathrm{d}\tau\mathrm{d}\sigma^2$$

$$=\frac{\Gamma(\alpha + \frac{1}{2})}{\Gamma(\alpha)}\sqrt{\frac{v}{2\pi\beta(1+v)}}\Big(1 + \frac{v(\pi - \gamma)^2}{2\beta(1+v)}\Big)^{-(\alpha+\frac{1}{2})}$$

$$=\text{St}\Big(\pi; \gamma, \frac{\beta(1+v)}{v\alpha}, 2\alpha\Big) \tag{7}$$

Among the posterior NIG parameters, $\gamma$ is the expected mean of the predictive distribution:

$$\hat{\pi} = \mathbb{E}_{p(\pi|x,\mathbf{p})}[\pi] = \int \pi p(\pi|x, \mathbf{p})\mathrm{d}\pi = \gamma \tag{8}$$

According to [22], the other three evidential NIG parameters $\nu$, $\alpha$, and $\beta$ accumulate as the observations increase. The updates of these posterior parameters work as follows:

$$\nu_N = \nu + N, \quad \gamma_N = \frac{\nu}{\nu_N}\gamma + \frac{1}{\nu_N}\sum_{N=1}^{N}\pi_n, \quad \alpha_N = \alpha + \frac{N}{2} \tag{9}$$

$$\beta_N = \beta + \frac{1}{2}\sum_{k=1}^{N}(\pi_n - \sum_{N=1}^{N}\frac{\pi_n}{N})^2 + \frac{N\nu}{2(\nu + N)}(\sum_{N=1}^{N}\frac{\pi_n}{N} - \gamma)^2 \tag{10}$$

As we can see, each observation impacts the confidence of the model through these three parameters. $\nu$ and $\alpha$ can be seen as the pseudo-counts and directly impact the model evidence i.e., quantify the confidence on the prior mean and the prediction of a target data sample, respectively. Furthermore,

a large $\beta$ leads to low confidence in the model's prediction, which implies a lack of evidence. Additionally, each observation increases the pseudo-count of $\nu$ by 1, and $\alpha$ by $\frac{1}{2}$. Thus, we obtain the final form of the overall evidence $\mathcal{E} = v + \frac{1}{2}\alpha + \frac{1}{\beta}$ and the evidence-based regularization term $L_{REG}$.

## C  Detailed Algorithms

We provide our proposed model and AL strategy as two main algorithms shown in Algorithms 1&2: the first one describes the entire AL process, while the second one describes the detailed training process of EMM.

---

**Algorithm 1** Active Learning with EMM Model (Outer Loop)

---

**Input** : Total number of AL rounds: $T$, AL batch size $|b_t|$
   Unlabeled pool: $S_U$, annotation method: $h : \mathbf{x} \to \mathbf{y}$
   Model at step $t$: $f_{\theta_t}(\mathbf{x})$, consisting of $e(\mathbf{x}), g_{\boldsymbol{\pi}}(\mathbf{x}), g_{\boldsymbol{\mu}}(\mathbf{x})$
   AL sampling strategy $\mathcal{A} : f_{\theta_t}(\mathbf{x}) \times S_U \to \mathbb{R}$,
   Learning objective $\mathcal{L}$: $f_{\theta_t}(\mathbf{x}) \times \mathbf{y} \to \mathbb{R}$,
**Output** : Annotated training dataset: $S_L$, final model $f_{\theta_T}$
Randomly select $S_L$                     // Balanced Split Method for Initial Training
**for** $t = 1$ **to** $T$ **do**
   1. Train preset components $\boldsymbol{\Theta}_0$ on $S_L$                // E-M: Bernoulli Mixture Training
   2. Compute preset weights $\boldsymbol{\Pi}_0$ with $\boldsymbol{\Theta}_0$                   // Linear Program Optimization
   3. Pre-train for weight coefficient predictions ($e()$ and $g_{\boldsymbol{\pi}}()$) to fit $\boldsymbol{\Pi}_0$        // $\mathcal{L}_{\texttt{EVID}}(\boldsymbol{\Pi}_0, \hat{\boldsymbol{\pi}})$
   4. Completely train EMM model ($e()$ and $g_{\boldsymbol{\pi}}(), g_{\boldsymbol{\mu}}()$) to fit $\mathbf{y}$ // Alternating $\mathcal{L}_{\texttt{EVID}}/\mathcal{L}_{\texttt{SoftMargin}}$
   5. Active sample $b_t$ from $S_U$ based on $\mathcal{A}$                    // Multi-source $\mathcal{A}_{\texttt{MSU}}(\mathbf{x})$
   6. Update the pool and training set $S_U = S_U \backslash b_t, S_L = S_L \cup b_t$   // Prepare for next round
**end**

---

---

**Algorithm 2** EMM Model Training (Inner Loop)

---

**Input** : Model: $f_{\theta_t}(\mathbf{x})$, consisting of $e(\mathbf{x}), g_{\boldsymbol{\pi}}(\mathbf{x}), g_{\boldsymbol{\mu}}(\mathbf{x})$
   Learning objective $\mathcal{L}$: $\mathcal{L} : f_{\theta_t}(\mathbf{x}) \times y \to \mathbb{R}$ (Types: $\mathcal{L}_{\texttt{EVID}}(\boldsymbol{\Pi}_0), \mathcal{L}_{\texttt{SoftMargin}}(\mathbf{y})$)
**Output** : Trained model $f'$, updated label clusters $\boldsymbol{\Theta}'_0$, predictions $\hat{\mathbf{y}}$
**for** $i = 1$ **to** $epochs_{pretrain}$ **do**
   Train $e()$ and $g_{\boldsymbol{\pi}}()$ using $\mathcal{L}_{\texttt{EVID}}(\boldsymbol{\Pi}_0)$

**end**
**for** $j = 1$ **to** $epochs_{train}$ **do**
   **for** $k = 1$ **to** $epochs_l$ **do**
      Freeze $e()$ and $g_{\boldsymbol{\pi}}()$
      Train $g_{\boldsymbol{\mu}}()$ using $\mathcal{L}_{\texttt{SoftMargin}}(\mathbf{y})$
      ($\boldsymbol{\Theta}(\mathbf{x})$ is predicted from $g_{\boldsymbol{\mu}}()$ for each point, $\boldsymbol{\Theta}'_0 = \boldsymbol{\Theta}_0 + \sum_n w_{new}\boldsymbol{\Theta}(\mathbf{x}_n)$)

   **end**
   **for** $l = 1$ **to** $epochs_{\pi}$ **do**
      Train $e()$ and $g_{\boldsymbol{\pi}}()$ using $\mathcal{L}_{\texttt{EVID}}(\boldsymbol{\Pi}_0)$
   **end**
**end**

---

## D  Experiment Details and Additional Results

### D.1  Experiment settings

Our experiments are performed on clusters with NVIDIA A6000 and NVIDIA A100 graphic cards and Intel Xeon Gold 6150 CPU processors. The code is implemented with PyTorch [24]. The runtime of the experiments varies depending on the size of the unlabeled pool. Compared to traditional

models, the evidential deep learning model takes a longer time to train. However, compared to Bayesian models or label ranking models, the inference time is significantly shorter.

For our main results, we pre-train the label clusters using an E-M algorithm and obtain initial optimal weights for the labeled training set using linear programming optimization. The evidential model is trained for around 5000 epochs to fit the weight coefficients $\pi$, and the joint training step is trained iteratively for 100 epochs in each round.

## D.2  Additional Baseline Comparison

Here we show additional AL comparison with two other baselines:

- **AUDI** uses a label ranking mechanism, where a dummy label is used to separate the positive and negative labels. Its sampling function is based on a modified cardinality inconsistency measure [14].
- **CS** uses a compressed sensing mechanism combined with GP predictions [31].

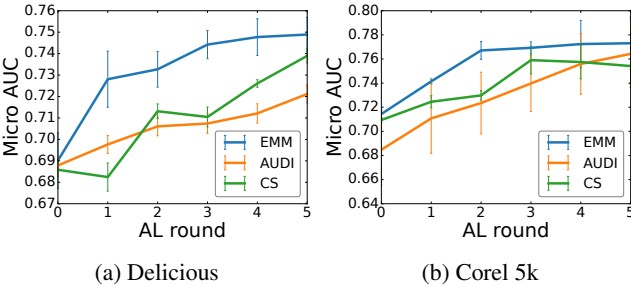

(a) Delicious          (b) Corel 5k

Figure 8: Additional active learning performances comparison on smaller-sized real-world datasets (the AUDI and CS baselines become computationally expensive on larger datasets)

From Figure 8 we can see that both baselines do not perform well at the low budget, while EMM still outperforms them. The implementation of AUDI and CS on BibTex and NUS-WIDE are computationally too demanding and we would expect similar behavior given the labeling budget.

Most recent multi-label works also focus on AUC optimization. So, we also add a discussion on the macro-AUC. For example, [36] achieves the state-of-the-art macro-AUC results on bibTex, Corel 5k and Delicious. In our work, due to the poor macro-AUC performance of some baselines, we mainly present micro-AUC results. However, here we provide the macro-AUC results using 80% training data (except for NUS-WIDE where we use 13,000 training samples because full-training is too expensive) compared to the methods in [36]. We can see that the macro-AUC is close on common datasets. Additionally, the AUC optimization method is orthogonal to our evidential model. We could incorporate AUC-based loss regularization into our joint-label training step. Due to the complexity, we leave these studies to future work. Instead, we focus on the AL improvements in this paper.

Table 3: Performance metrics of EMM across datasets

| EMM | Delicious | Corel 5k | bibTex | NUS-WIDE |
|---|---|---|---|---|
| **micro-AUC** | 0.8021 | 0.8063 | 0.8669 | 0.8625 |
| **macro-AUC** | 0.7256 | 0.6613 | 0.8153 | 0.6396 |

Table 4: Repetition of EMM metrics across datasets

| EMM | Delicious | Corel 5k | bibTex | NUS-WIDE |
|---|---|---|---|---|
| **micro-AUC** | 0.8021 | 0.8063 | 0.8669 | 0.8625 |
| **macro-AUC** | 0.7256 | 0.6613 | 0.8153 | 0.6396 |

Table 5: AUC-surrogate performance on selected datasets

| AUC-surrogate | Delicious | Corel 5k | bibTex |
|:---:|:---:|:---:|:---:|
| $A_{u1}$ | 0.7633 | 0.6645 | 0.8693 |
| $A_{u2}$ | 0.8044 | 0.5703 | 0.9299 |

## D.3 Additional Ablation Study

In this set of experiments, we show results using more configurations of $\lambda$ and $\eta$. As explained in the main paper, these balancing parameters work well with a moderately small value. With $\lambda$ and $\eta$ both around 0.1 to 0.01, we are able to obtain stable AL results. However, if the values are set too large, the performance may degrade.

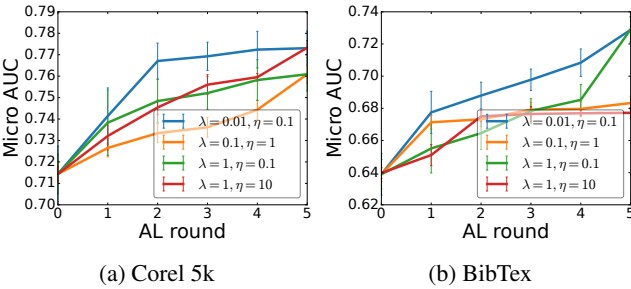

(a) Corel 5k                    (b) BibTex

Figure 9: Additional ablation study on balancing parameters: different values

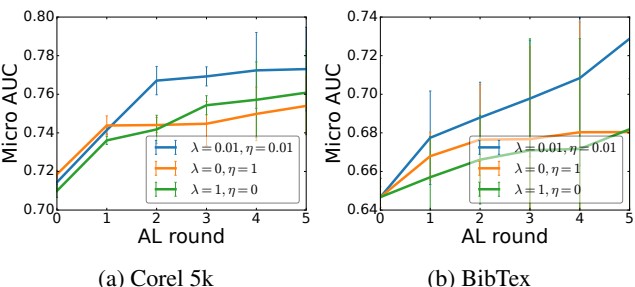

(a) Corel 5k                    (b) BibTex

Figure 10: Additional ablation study on balancing parameters: single uncertainty source

When we set $\lambda = 0, \eta = 1$ or $\lambda = 1, \eta = 0$, we get the combination of $\mathcal{A}_{\pi_k}$ and a single source of label uncertainty $\mathcal{A}_\mu$ or $\mathcal{A}_{\hat{y}}$. As we can see, the combination of all three works the best. In Figure 11, we show the comparison between different $K$ values. As we can see, extremely low number of clusters is not sufficient for achieving good model performance, while a large number is much more costly and also suffers from overfitting. The latter problem might harm the AL sampling more as we see the $K = 10$ case performs even worse than $K = 3$.

## D.4 Additional Uncertainty Analysis

We can also evaluate the quality of uncertainty estimation using true labels of the pool samples after AL experiments. Here, we present an example of the rare-label uncertainty being captured by both the cluster difference prediction and the covariance of the predicted labels. From Table 6 (results obtained on BibTex), we can see that similar to the synthetic data case, the uncertainty metric captures the rare labels well, although the difference is smaller because there are more labels in total.

# E    Limitations, Future Works and Broader Impact

While EMM offers significant advancements in multi-label active learning, there are some limitations to consider. First, the model's complexity and computational requirements may pose challenges when

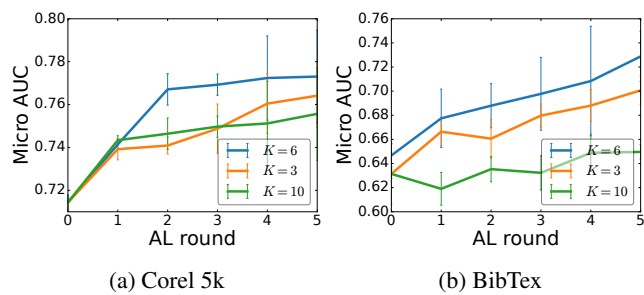

(a) Corel 5k                    (b) BibTex

Figure 11: Additional ablation study on number of clusters $K$

Table 6: The relationship between average uncertainty scores, label cardinality, and rare labels

|  | $|\mathbf{y}|<3$ | $|\mathbf{y}| \geq 3$ | $\mathbf{y}_{L1} = 1$ | $\mathbf{y}_{L1} = 0$ |
|---|---|---|---|---|
| Average $\mathcal{A}_{\pi_k}(\mathbf{x})$ | 3.886 | 3.062 | 3.834 | 3.196 |
| Average $\mathcal{A}_{\hat{\mathbf{y}}}(\mathbf{x})$ | 0.021 | 0.064 | 0.018 | 0.013 |

applied to extremely large datasets or in real-time applications. Second, we have not integrated the Bayesian selection for $K$ in EMM, making the choice of $K$ crucial. Our future works include a more adaptive model structure that allows a dynamic $K$ as the number of label clusters, along with a more lightweight training and adaptation process of the model. We also look to integrate adaptive testing and validation approaches for the AL process.

EMM has the potential to significantly impact various fields that rely on multi-label classification. In domains such as healthcare, bioinformatics, and environmental science, where data labeling is often expensive and time-consuming, our model can optimize the use of limited labeling resources, leading to more efficient and accurate predictions. This can accelerate research and development in these fields by enabling the discovery of new patterns and correlations. Moreover, the model's sophisticated uncertainty quantification can improve decision-making processes in critical applications, where understanding the confidence and reliability of predictions is crucial. However, the implications of label clusters need to be carefully considered in applications.

# F   Source Code

https://github.com/ritmininglab/EMM.git

