# OpenReview forum: "Evidential Mixture Machines: Deciphering Multi-Label Correlations for Active Learning Sensitivity"
_NeurIPS.cc/2024/Conference — NeurIPS 2024 poster_

### Official Review · Reviewer_9WcB · 2024-07-06

**Soundness:** 2
**Presentation:** 2
**Contribution:** 2
**Rating:** 5
**Confidence:** 3

**Summary:**

The paper introduces an active learning approach based on Evidential Mixture Machines (EMM) that compress the large, sparse label space of multi-label problems into a more manageable weight coefficient space. This approach combines mixtures of Bernoulli with a Deep Evidentiary models, and it leverages multiple sources of uncertainty (i.e., evidential uncertainty and the predicted label embedding covariance) to improve the active sample selection,

**Strengths:**

The paper introduces a novel approach to multi-class active learning. The approach appears to be original, and it improves over existing approaches on four domains. Due to reasons explained in the next section, the quality and significance of this work are difficult to judge in the paper's current form.

**Weaknesses:**

The paper could be improved on two main dimensions. First, the authors could make their case more compelling and intuitive by (i) introducing a motivating real-world example, and (ii) adding a section with an intuitive running examples that explains in layman terms the contribution of each of EMM's components, together with the synergy among them. Second, the experimental section should be expanded with a comprehensive analysis of the 4 domains in section 4.2 and their properties (e.g., the distribution of the multi-labels, so that one can grasp the challenges in each dataset), and by adding to each graph the best-known performance on that dataset in via fully supervised training. Without this last bit of information, it is impossible to judge the impact of this approach: the correct question is how much can AL reduce the need for labeled data while maintaining SOTA performance, rather than "how well can I do with X examples chosen via AL". For example, multiple algorithms reach 95+% accuracy on BIBTEX; how should we compare such results with 73% Micro AUC? To allow the reader/reviewer to estimate the significance of EMM, the authors should provide and discuss these numbers.

**Questions:**

- please provide a reference for the claim in. lines 52-53
- line 61 - please explain why a single pass is sufficient
- line 321 - the details are not presented in. Appendix D
- lines 329-332 - instead on percentages from the original  data set, use absolute numbers; also justify the reason for choosing those particular numbers

---

> ### Author Rebuttal · Authors · 2024-08-07
>
> **Q1: (i) Introducing a motivating real-world example, and (ii) adding a section with an intuitive running example that explains in layman terms the contribution of each of EMM's components, together with the synergy among them.**
>
>
> Thank you for this great suggestion! For the motivation real-world example, let's consider AI facilitated medical diagnosis, where the goal is to predict the likelihood of various diseases for patients based on their medical records. Each patient can have multiple possible conditions (classes). Traditional methods may struggle due to the rarity and interdependence of certain conditions. For example, diseases like rare genetic disorders occur infrequently, making it difficult for models to learn effectively from limited labeled data. Moreover, some conditions often co-occur or influence each other, such as diabetes and high blood pressure. The proposed EMM addresses these challenges more effectively by leveraging label correlations and predicts rare conditions by utilizing a mixture model approach that considers the dependencies among diseases.
>
> As for the contribution of each component regarding question (ii), the active learning strategy employed by EMM ensures that the most informative patient records are selected for labeling, optimizing the use of limited medical resources. This approach not only improves prediction accuracy but also provides robust uncertainty estimates, which are crucial for making reliable medical decisions. Furthermore, evidential learning plays a critical role in this process by providing robust uncertainty estimates for the predictions. This is particularly important in healthcare and other knowledge rich domains, where the cost of incorrect predictions can be high. Evidential learning allows the model to quantify its confidence in each prediction, enabling the active learning component to prioritize labeling the most uncertain and potentially informative records. This ensures that the model not only improves prediction accuracy but also provides reliable and interpretable predictions as active learning goes. Following the reviewer's suggestion, we will incorporate this example into our paper to make the proposed approach more intuitive and relatable to readers while highlighting the contribution of each key component.
>
> **Q2: How should we compare the results with full training?**
>
> We appreciate the reviewer's insightful comment. Indeed, it is important to demonstrate how active learning (AL) with EMM reduces the need for labeled data while maintaining state-of-the-art (SOTA) performance. The complexity of SOTA metrics in multi-label problems means that accuracy can vary significantly depending on whether class imbalance is considered and how decision-making strategies are applied. Specifically, the 95+\% accuracy mentioned by the reviewer may not be directly comparable due to different decision thresholds used by various methods. We chose the AUC metric because it focuses solely on prediction quality, independent of decision thresholds, providing a fairer assessment of multi-label classification performance.
>
> Most recent multi-label works also focus on AUC optimization. For example, [1] shows SOTA macro-AUC results on bibTex, Corel 5k and Delicious. In our paper, due to the poor macro-AUC performance of some baselines, we mainly present micro-AUC results. However, here we provide the macro-AUC results using 80\% training data (except for NUS-WIDE where we use 13000 training samples as full-training is too expensive) compared to the methods in [1]. We can see that the macro-AUC is close on common datasets. Additionally, the AUC optimization method is orthogonal to our evidential model. We could incorporate AUC-based loss regularization into our joint-label training step. Due to the complexity, we leave these studies to future works. Instead, we focus on the AL improvements in this paper.
>
> |   EMM    | Delicious | Corel 5k | bibTex   | NUS-WIDE |
> |----------|-----------|----------|----------|----------|
> | micro-AUC| 0.8021    | 0.8063   | 0.8669   | 0.8625   |
> | macro-AUC| 0.7256    | 0.6613   | 0.8153   | 0.6396   |
>
> | AUC-surrogate | Delicious | Corel 5k | bibTex   |
> |---------------|-----------|----------|----------|
> | $A^{u1}$      | 0.7633    | 0.6645   | 0.8693   |
> | $A^{u2}$      | 0.8044    | 0.5703   | 0.9299   |
>
> **Q3: Other questions**
>
> For the challenges of scalability and predictive ability faced by the Gaussian process due to the explosion of data size, see the survey paper [2]. While the fact that sparse Gaussian Process models are limited in their predictive ability is supported by [3]. We would like to add those references to the next version of the paper.
>
>
> In line 61, we refer to the prediction phase of deep learning models, where only a single forward pass is required to generate predictions.
>
> We apologize for missing the dataset summaries in Appendix D. We will include them along with the absolute samples used in the experiments in the next version of the paper. In addition, we want to assure the reviewer that all the datasets used in our experiments are publicly available, and our preprocessing steps follow standardized procedures. Therefore, the missing of this information in the current version does not critically affect the credibility of our results.
>
> **References**
>
> [1] Wu, Guoqiang, Chongxuan Li, and Yilong Yin. "Towards understanding generalization of macro-AUC in multi-label learning." International Conference on Machine Learning. PMLR, 2023.
>
> [2] Liu, Haitao, et al. "When Gaussian process meets big data: A review of scalable GPs." IEEE transactions on neural networks and learning systems 31.11 (2020): 4405-4423.
>
> [3] Lederer, Armin, Jonas Umlauft, and Sandra Hirche. "Uniform error bounds for Gaussian process regression with application to safe control." Advances in Neural Information Processing Systems 32 (2019).

---

> > ### Comment · Reviewer_9WcB · 2024-08-08
> > **Many thanks to the authors for the detailed answers. After considering both the other reviews and the rebuttals, I maintain the original rating.**
> >
> > Many thanks to the authors for the detailed answers. After considering both the other reviews and the rebuttals, I maintain the original rating.

---

> > > ### Author Response · Authors · 2024-08-08
> > >
> > > Thank you for reading our rebuttal and maintaining a positive rating. Again, we appreciate your constructive feedback and will incorporate the suggestions when revising our paper.

---

### Official Review · Reviewer_fLE1 · 2024-07-08

**Soundness:** 2
**Presentation:** 2
**Contribution:** 2
**Rating:** 5
**Confidence:** 2

**Summary:**

This manuscript investigates multi-label active learning, a critical issue in contemporary machine learning. To address this challenge, the authors introduce a novel evidential mixture machines (EMM) model, which provides an uncertainty-aware connection from input features to the predicted coefficients and components. The performance of the developed method is demonstrated through simulations on synthetic and real-world datasets.

**Strengths:**

The authors argue that the EMM model can produce a richer multi-source uncertainty metric than simple uncertainty scores and enhance prediction accuracy. The model sounds interesting.

**Weaknesses:**

However, there are several typographical errors in the manuscript, and the authors should thoroughly review and revise the text. The authors should improve the presentation. Additionally, more details about the model should be provided.

**Questions:**

Specific comments and questions:
1.	How do the authors determine the parameters in the student-t distribution?
2.	In Section 4.1, the authors mention the large or small value of the weight. How does the outcome change with different weight values? Is the performance of the EMM model closely related to the weight? If so, how is the weight determined in different scenarios without prior information about the labels?
3.	In Figure 6, although EMM has a significant advantage on rarer labels, several high orange bars remain in Figure 6 (b). Could the authors explain this? Additionally, please add numerical labels to the y-axis below the API=0 axis.
4.	What is the computational complexity of the EMM method compared to state-of-the-art (SOTA) methods, and how does it scale with the size of the dataset?
5.	Please explain how this method can be employed in practical applications and what its strengths are compared to other SOTA methods (beyond accuracy).

**Limitations:**

The authors have adequately addressed the limitations of their work.

---

> ### Author Rebuttal · Authors · 2024-08-07
>
> **Q1: How do the authors determine the parameters in the student-t distribution?**
>
>  The parameters are a crucial component of the Bayesian nature of the evidential model. The parameters for the student-t distribution ($\pi,\gamma,\frac{\beta(1+\nu)}{\nu\alpha},2\alpha$) are all obtained from network outputs, as specified in Line 163 and the following parts. More details of the interpretation are given in Appendix B.
>
> **Q2: How does the outcome change with different weight values?**
>
> If the question refers to the coefficient weights $\pi$, they are obtained from the evidential network predictions. A larger $\pi_k$ will indicate that the sample can be better explained by label cluster $k$. This is how we connect the data sample to the Bernoulli mixture label clusters. The initial label clusters are learned through E-M on training labels, and the weights are learned throught evidenital training. If the question refers to the weights of the uncertainty components, they are explained in Section 3.3 and we provide an ablation study in Section 4.2 and Appendix D.4.
>
> **Q3: Several high orange bars remain in Figure 6 (b).**
>
> Thank you for the suggestion. We will improve the figure in the revised paper. According to [1], the GP-B2M has a good performance on rare labels. We show that we have improved performance on most datasets (also in Figure 9). The Delicious dataset is a more imbalanced dataset, so it might be more difficult to perform well on all rare labels.
>
> **Q4: What is the computational complexity of the EMM method?**
>
> Compared to the GP-B2M model which requires a $\mathcal{O}(N^3)$ complexity during prediction time (due to the inference process of GP), we have a linear complexity in the number of samples. During training, the complexity is at the same scale as a standard evidential neural network.
>
> **Q5: Please explain how this method can be employed in practical applications and what its strengths are compared to other SOTA methods (beyond accuracy).**
>
>
> First, EMM is an end-to-end multi-label model. Thus, in real-world applications, EMM could be utilized in the same way as other models, such as binary relevance machines and label ranking models. Then, it can be applied to the AL task as demonstrated in real-world experiments, where the fine-grained uncertainty could be utilized for active sampling. The main advantages of the EMM model include the scalability as a deep evidential model compared to traditional Bayesian models and the uncertainty information compared to standard neural networks. For a good example of the employment of EMM, please also refer to the response to Q1 from Reviewer 9WcB.
>
> [1] Shi et al. "A gaussian process-bayesian bernoulli mixture model for multi-label active learning." NeurIPS 2021.

---

> > ### Author Response · Authors · 2024-08-12
> >
> > We sincerely appreciate the time you dedicated to reviewing our paper. We have clarified the parameters and variables settings in our model.  We have also provided additional descriptions of the characteristics of our model. We hope that our response adequately addresses your inquiries. Should you have any further questions or require additional clarifications, please let us know.

---

> > ### Comment · Reviewer_fLE1 · 2024-08-13
> >
> > Thanks for the authors' response. The authors have addressed most of my concerns. I would like to maintain my score.

---

> > > ### Author Response · Authors · 2024-08-13
> > >
> > > Thank you for reading our rebuttal and confirming that it has addressed most of your concerns. We appreciate that you maintain the positive rating on the paper!

---

### Official Review · Reviewer_FbZQ · 2024-07-11

**Soundness:** 3
**Presentation:** 3
**Contribution:** 3
**Rating:** 6
**Confidence:** 2

**Summary:**

This paper focused on multi-label classification problems in active learning settings, where the label relationship, especially for rare labels, is hard to learn. The authors proposed an Evidential Mixture Machine (EMM) that combines the mixture of Bernoulli with a deep evidential model, which allows joint learning of the weight coefficients pseudo label counts to model the proxy label distribution for each instance. This method allows quick inference time and twofold uncertainty measurement from the evidential posterior parameters and the predicted variability in final labels. Extensive experiments on real-world and synthesised datasets certified the stated advantages of the proposed methods.

**Strengths:**

- The authors clearly stated the motivation for the proposed model, where in a multi-label classification problem, it is hard to learn the dependencies between labels or the correlation of rare labels, which will be amplified in an active learning setting.
- The authors have provided a thorough critique of current research methods, highlighting their shortcomings. Their proposed method effectively addresses the limitations of existing CBM and GP-B2M based methods.
- The idea of combining different uncertainty measurement sources is appealing, and the results show that the strategy is beneficial comparing to the strategy based only on entropy .
- Comprehensive experiments and ablation study which support the claims

**Weaknesses:**

Few typos and confusion:
- line 103 "reuarization"
- lines 339-341 stated that EMM model outperforms the others in the initial round, which only showed in Figure 3 (c) Bibtex. Also, I can't see "the GP-B2M model may also have good performance at the starting point" in lines 341-342

**Questions:**

- Figure 7 in Appendix D.2 is ambiguous about the label relationships. Why is the Geo-based labels pointing to L1 rather than the Non-geo-based labels? Also, the arrows from L3 to L4 doesn't show the not relation.
- From the results in Figure 3, the proposed active learning strategy doesn't have noticeable advantage than the EMM-entropy in (d) and (a). Does it indicate that for some dataset, the proposed active learning strategy doesn't add much information about uncertainty compared to entropy?

**Limitations:**

The authors have adequately addressed the limitations and potential societal impact of their work.

---

> ### Author Rebuttal · Authors · 2024-08-07
>
> Thank you for providing constructive comments/suggestions. Below, we provide the response to the questions and comments.
>
>
> **Q1: Typos and confusion.**
>
> Thank you for pointing out the typos. We will correct them in the revised paper. The EMM model does perform better on Corel 5k and BibTex in the initial round as can be seen in Figures 3(b) and (c). The EMM-entropy baseline should share the same initial performance as it also uses EMM as the base model. However, we made a plotting error by plotting the EMM-fixed (from the ablation study) instead of EMM-entropy in Figure 3(c). We will correct this in the revised paper (please see the pdf file under general response for the corrected figure). The description in line 341 is inaccurate. In our experiments, the label ranking model or the BRM model may have a good starting point compared to other baselines as shown in Figure 3 (a) and (d).
>
> **Q2: Confusion about Figure 7.**
>
> We apologize for the confusion. In Figure 7, we meant to show the generation process of the Non-geo-based labels, not the actual logical relationships between them. L1 is randomly sampled from data points with certain Geo-based labels. L2, L3 and L4 are than generated based on the process in the description in Section 4.1. We will improve the quality of the figure and clarify this in the figure caption and Appendix D.2.
>
> **Q3: Comparison with EMM-entropy.**
>
> Thank you for raising an interesting point about the potential lower information advantage on certain datasets. Regarding the performance comparison, because the EMM-entropy baseline also uses the proposed EMM model, the difference only comes from the sampling method. Thus, the AL curves can be close on some datasets. The assumption that the information from other components has less impact than entropy is quite plausible. As we can see from the ablation study, the performance change from the balancing parameters in the sampling function $\lambda$ and $\eta$ is less obvious on Delicious and NUS-WIDE than on Corel 5k and BibTex. To better support this finding, we compare the variance of the uncertainty scores (normalized over 500 pool samples) on these datasets. From the results below, we can see that the variance of the weight coefficient uncertainty is much higher compared to the other two components on Delicious and NUS-WIDE. Thus, the weight coefficient uncertainty might have a larger impact. The EMM-entropy uses the entropy criterion from the weight coefficient side, thus might perform closer to the integrated EMM method on these datasets. We agree that this could be a very interesting factor in real-world AL applications and will further study the problem in future works.
>
> | Var($\mathcal{A}$)      | Delicious  | Corel 5k  | bibTex    | NUS-WIDE  |
> |-------------------------|------------|-----------|-----------|-----------|
> | Var($\mathcal{A}_\pi$)  | 1.219e-3   | 6.074e-4  | 7.683e-4  | 1.984e-3  |
> | Var($\mathcal{A}_\theta$)| 2.561e-5  | 1.004e-4  | 2.635e-5  | 9.805e-5  |
> | Var($\mathcal{A}_y$)    | 4.473e-5   | 1.028e-4  | 4.763e-5  | 1.439e-4  |

---

> > ### Comment · Reviewer_FbZQ · 2024-08-11
> >
> > I appreciate the authors' response to my review and their efforts to address my concerns. After carefully reviewing the feedback from the other reviewers, I am inclined to maintain my original score but remain open to reconsideration during the discussion stage with the other reviewers and the Area Chair.

---

> > > ### Author Response · Authors · 2024-08-12
> > >
> > > Thank you for reading our rebuttal and maintaining a positive rating. We will incorporate your feedback into the revised paper and are more than happy to answer any further questions that you may have.

---

### Official Review · Reviewer_dBCp · 2024-07-13

**Soundness:** 3
**Presentation:** 3
**Contribution:** 2
**Rating:** 5
**Confidence:** 4

**Summary:**

This paper introduces the Evidential Mixture Machines (EMM) model, which addresses the multi-label active learning tasks, particularly in rare-class scenarios. EMM uses a mixture of Bernoulli distributions to capture label correlations and uses evidential learning to quantify uncertainties for more informed active sample selection. The model dynamically updates label clusters using proxy pseudo counts.

**Strengths:**

1. The usage of a mixture of Bernoulli dist. is interesting, By using it, the model effectively captures label correlations, which is crucial for multi-label classification where labels are often interdependent.
2. The usage of evidential learning is also interesting, it can be used for predicting weight coefficients and provides fine-grained uncertainty quantification. This allows the model to differentiate between epistemic and aleatoric uncertainty.
3. The integration of multi-source uncertainty in the active learning strategy promotes active learning performance (shown in the ablation study), it improves the selection of informative samples, leading to better performance with fewer labeled instances.

**Weaknesses:**

I have 2 concerns:
1. Is the model robust to outliers? In real-life data scenarios, the existence of outliers is normal. Outliers can cause the proposed model to learn incorrect or spurious correlations between labels, (label noise issues also have a similar effect). If outliers have unusual combinations of labels, they can distort the representation of the label clusters.

2. Compared with the GP-B2M paper's experiment, there are some inconsistencies, for example, on Bibtex and Corel5K datasets, GP-B2M performs significantly better than CVIRS in GP-B2M paper, but performs similarly on this manuscript, can the author discuss more about it?

**Questions:**

Please see the weakness part.

**Limitations:**

The author has already addressed potential biases and boarder impact in their appendix.

---

> ### Author Rebuttal · Authors · 2024-08-07
>
> Thank you for providing constructive comments/suggestions. Below, we provide the response to the questions and comments.
>
> **Q1: Is the model robust to outliers?**
>
> Thank you for the suggestion for this interesting challenge. We agree that being robust to outliers is an important challenge in real-world applications. In active learning, the scarcity of labels may cause the model to be more brittle. Although we do not deal with outliers directly in the proposed method, there is potential for AL with the existence of outliers using our fine-grained uncertainty analysis. For example, we could derive the predicted vacuity of data samples, which indicates how little support the evidential model has for these samples, to identify potential outliers. Better utilizing the EMM model for AL with outliers could be an important future direction of this work. We will add the discussion in the revised paper as well.
>
> **Q2: Different performance using the CVIRS baseline from previous papers.**
>
> In our experiments, the performances are slightly different because we use a larger AL batch size. If we compare the AUC after selecting 500 samples, it is similar to those in the previous paper. We will further clarify the setting in the revised paper. In the GP-B2M paper, the same baselines were implemented using either BRM or the GP-B2M model. We adapted the CVIRS baseline using the EMM model but the results are inferior to those of EMM-entropy. To be consistent, we present the results using the binary relevance machines (BRM) model.

---

> > ### Comment · Reviewer_dBCp · 2024-08-11
> > **response**
> >
> > The answer was not deep enough and I decided to keep the grade.

---

> > > ### Author Response · Authors · 2024-08-12
> > >
> > > Dear Reviewer dBCp,
> > >
> > > Thank you for reading our rebuttal! The outlier problem in the multi-label setting is a relatively less explored topic. One potential reason is that outliers are less well-defined in multi-label problems. Even in existing multi-label datasets, the imbalance issue among labels is already severe. Also, as we explained in our motivation section, there are many underlying correlations between these labels that might be difficult to define. Thus, the design of the outliers problem itself is challenging in multi-label classification. Coupled with active learning, where labels are scarce, the problem may become more prominent. As for how our proposed method can benefit the topic, we can extend the point in our previous response about using our fine-grained uncertainty analysis to improve outlier detection. In particular, for labeled data samples where the model exhibits a relatively low epistemic uncertainty (as defined in Eq. 4), which implies that the model has a decent knowledge of the sample, if the model prediction deviates significantly from the human-provided label, it could indicate a potential outlier. In this case, we may ask the annotator to re-check the label and provide a new label if a mistake is made. Such an *active relabeling* process can be integrated into an outer active learning loop to minimize the impact of outliers on the overall learning process. While this could be an interesting research direction, thoroughly exploring this topic is out of the scope of our current work. We will extend our future work section by adding a detailed discussion on this topic.

---

### Author Rebuttal · Authors · 2024-08-07

**General Response**

We would like to thank all reviewers for spending time to review our paper and providing constructive comments/suggestions. Below, we summarize some of our major responses:

- *Our contributions and how EMM could be used*: As stated in lines 83-87, our contributions include the novel integration of evidential learning and multi-label model in EMM and the corresponding active sampling method accompanying the model. Since reviewers asked about the usage of EMM, we clarify that it is first a multi-label model that can be readily used. Then, it is suitable for AL given the fine-grained uncertainty information. More detailed real-world examples are given in the specific responses.

- *Baselines and ablation studies*: Below in the specific responses, we clarify the confusion about the baselines and ablation studies and correct some minor errors.

- *Typos and details*: We thank all reviewers for pointing out the typos and missing details. We will correct those in the revised paper and provide the details as requested.

---

### Decision · Program_Chairs · 2024-09-25

**Decision:**

Accept (poster)

**Comment:**

Reviewers expressed mixed but generally positive views on the paper. While the novel approach to multi-label active learning was appreciated, particularly the use of Bernoulli mixtures and evidential learning, there were concerns about robustness to outliers and the clarity of the presentation. Some reviewers noted that while the model performs well, the paper could benefit from more intuitive explanations and improved experimental analysis.

As a meta-reviewer, I noticed that the paper does not explain in sufficient detail how the synthetic data were generated. There are explanations in Section 4.1 and Supplementary D.2, but neither is sufficiently detailed, making replication difficult based on the text alone. There is a link for anonymously hosted source code, but the link had expired by the time I looked at it. Despite these issues, the authors’ rebuttal addressed most concerns, and with some refinements, the paper is suitable for acceptance.